

# AI-based automated breast cancer segmentation in ultrasound imaging based on Attention Gated Multi ResU-Net

Ting Ding[1,2], Kaimai Shi[3], Zhaoyan Pan[4] and Cheng Ding[5]

[1] School of Earth Science, East China University of Technology, Nanhang, JiangXi, China
[2] Urumqi Comprehensive Survey Center on Natural Resources, Urumq, XinJiang, China
[3] School of Physics, Georgia Institution of Technology, Atlanta, GA, USA
[4] School of Energy Power Engineering, Xian Jiaotong University, Xian, China
[5] Biomedical Engineering, Georgia Institute of Technology, Atlanta, GA, United States of America

## ABSTRACT

Breast cancer is a leading cause of death among women worldwide, making early detection and diagnosis critical for effective treatment and improved patient outcomes. Ultrasound imaging is a common diagnostic tool for breast cancer, but interpreting ultrasound images can be challenging due to the complexity of breast tissue and the variability of image quality. This study proposed an Attention Gated Multi ResU-Net model for medical image segmentation tasks, that has shown promising results for breast cancer ultrasound image segmentation. The model's multi-scale feature extraction and attention-gating mechanism enable it to accurately identify and segment areas of abnormality in the breast tissue, such as masses, cysts, and calcifications. The model's quantitative test showed an adequate degree of agreement with expert manual annotations, demonstrating its potential for improving early identification and diagnosis of breast cancer. The model's multi-scale feature extraction and attention-gating mechanism enable it to accurately identify and segment areas of abnormality in the breast tissue, such as masses, cysts, and calcifications, achieving a Dice coefficient of 0.93, sensitivity of 93%, and specificity of 99%. These results underscore the model's high precision and reliability in medical image analysis.

## INTRODUCTION

Breast cancer is the most frequent cancer in women worldwide and is the second most significant cause of cancer mortality in women after lung cancer. Breast cancer makes up for around 25% of all malignancies in women, according to the World Health Organization (WHO). In 2018, it is anticipated that 2.1 million new breast cancer cases will be diagnosed and 627,000 people will die from the disease in 2021–2022 (*DeSantis et al., 2019*). Breast cancer becomes more frequent with age, and it is most typically diagnosed in women over 50. It can, however, develop in younger women or even males, but it is considerably less

Corresponding author
Cheng Ding, cheng.ding2@emory.edu

prevalent in men. Several variables increase the chance of breast cancer, including genetics, hormonal effects, lifestyle factors such as nutrition and physical exercise, and environmental exposures. Depending on the kind and stage of the disease, clinical treatment for breast cancer may involve surgery, chemotherapy, radiation, and hormonal treatment. Breast cancer survival rates have increased due to advancements in therapy, and early identification through usual self and radiography is critical for improving results (*Ahmad, 2019*).

Ultrasound imaging is a non-invasive diagnostic procedure that produces scans of the body's interior using high-frequency sound pulses. Ultrasound is a systematic method for seeing the breast and can be helpful in the early detection of breast cancer. Traditionally, a professional healthcare expert, such as a radiologist or a sonographer, uses an ultrasound machine to see the breast tissue and check for irregularities (*Xu et al., 2019*). It is crucial to remember that the efficiency of breast cancer identification *via* ultrasound is based on the doctor's expertise and competence, as well as the reliability of the technology. For various reasons, clinicians may find it difficult to screen for breast cancer using ultrasonography manually. The challenges include human error, subjectivity, limited precision, and time consumption. As the volume of information grows, interpreting a large quantity of data becomes more difficult. As a result, a computer-aided diagnostic (CAD) algorithm is necessary to aid doctors and specialists in rapidly diagnosing these dangerous diseases to preserve valuable human lives (*Shah et al., 2022*; *Zhang et al., 2021*).

Artificial intelligence (AI) can potentially transform the detection and treatment of breast cancer (*Sheth & Giger, 2020*). AI systems can examine enormous volumes of data and recognize trends that humans may find difficult to recognize. It can assist in enhancing the precision and accuracy of breast cancer diagnosis, especially when combined with additional imaging modalities like radiography or ultrasonography. It has the potential to automate specific processes and lessen the effort for healthcare practitioners, allowing them to devote more time to more challenging and higher-level duties (*Sechopoulos, Teuwen & Mann, 2021*). It can help improve efficiency and productivity, particularly in situations with a high volume of images or cases to review. The algorithms can examine imaging data and detect anomalies suggestive of breast cancer before they are seen on regular imaging tests. It can result in early identification and diagnosis, leading to better results.

Machine learning and deep neural networks can significantly affect breast cancer diagnosis and segmentation in image analysis. Algorithms for machine learning are computer software that can learn from information without being specifically designed. They can be programmed to spot trends and make predictions or conclusions based on their facts. Machine learning (ML) algorithms can be trained to assess imaging data and identify anomalies that could be symptomatic of malignancy in breast cancer diagnosis and localization. Deep learning (DL) algorithms are machine learning algorithms inspired by the human brain's anatomy and functions. They can study enormous volumes of data, learn to detect patterns and make judgments based on that data. Deep learning algorithms may be trained to assess imaging data and identify particular areas of the breast that may be malignant in the context of breast cancer diagnosis and segmentation (*Shah & Kang, 2023*; *Houssein et al., 2021*). ML and DL are used to automate the identification and segmentation of breast cancer in image analysis, with the potential to increase the efficiency and accuracy

of these operations dramatically. However, it is crucial to emphasize that these methods are still in their early stages, and additional study is needed to grasp their potential and limits in this area properly.

Deep learning, particularly neural networks, has gained popularity recently due to its ability to perform well on various tasks. Among the key reasons for its popularity is deep learning models' capacity to autonomously learn characteristics and interpretations from information, allowing them to enhance their ability to perform various activities. Furthermore, the availability of massive amounts of data and computing resources has enabled the training of massive and sophisticated models capable of achieving state-of-the-art results on various tasks (*Shah et al., 2024*). DL is a powerful method that employs multiple-layered artificial neural networks to discover patterns and characteristics from vast volumes of data. It has been used effectively across various fields and sectors, including machine vision: natural language processing, healthcare, self-driving cars, robotics, finance, speech recognition, gaming, and more (*Esteva et al., 2021*; *Shah et al., 2022*; *Hassaballah & Awad, 2020*). Consequently, Accessibility to a wide range of samples is critical for training and evaluating DL algorithms in healthcare. However, gathering and sharing medical information remains challenging due to privacy considerations. The healthcare business deals with sensitive patient information such as personal identity, medical history, and test results, which must be safeguarded to comply with requirements (*Baccouche et al., 2021*).

A DL model can be used as an auxiliary system by radiologists and doctors for quick and efficient segmentation for timing purposes. This study proposes an attention-gated multi-resU-Net model to identify and segment malignancy in breast ultrasound data. The scans will be evaluated using a combination of convolutional neural networks (CNNs) and attention processes. The U-Net architecture is a form of CNN often used for image segmentation, and the attention algorithm allows the network to concentrate on certain portions of the image that are particularly significant for the goal of detecting malignancies. The model would be trained using a dataset of breast ultrasound pictures annotated with the existence of malignant tumors. After training, the network can be used to assess new scans and segment the existence of malignancy, focusing more on the problematic area to improve its capacity to identify malignancy.

The following are the primary contributions of this study:

- The Attention Gated Multi ResU-Net model is evaluated for breast cancer ultrasound data segmentation: We assessed the model's segmentation performance in terms of Dice score, sensitivity, specificity, and AUC values and showed that it performed well in recognizing tumor areas.
- Assessment of the model's ability to extract features: We tested the model's capacity to extract critical aspects of tumor locations and observed it to be successful in recognizing these characteristics.
- We evaluated the performance of our proposed model to the performance of other state-of-the-art segmentation architectures and observed that it outperformed these models in terms of segmentation accuracy.

- Possible clinical applications: Using the Attention Gated Multi ResU-Net model to segment breast cancer ultrasound pictures accurately might potentially increase the accuracy of breast cancer diagnosis and treatment planning, leading to better patient outcomes.

## RELATED WORKS

Breast cancer diagnosis has evolved significantly by incorporating ML and DL algorithms into multiple medical imaging modalities. Thermography, magnetic resonance imaging (MRI), mammography, and ultrasound imaging have particular advantages for early diagnosis and detection, leading to more individualized and efficient diagnosis. Innovative computer methods that increase the accuracy of image analysis, segmentation, and classification have improved these technologies. This section summarizes significant research and advancements in various fields, demonstrating how AI has revolutionized conventional diagnostic approaches and providing information on the cutting-edge approaches currently used in breast cancer diagnosis.

### Ultrasound imaging

*Ragab et al. (2022)* proposed the Ensemble Deep-Learning-Enabled Clinical Decision Support System for Breast Cancer Diagnosis and Classification. They utilized ultrasound scans for breast cancer detection, employing Kapur's entropy and the Chaotic Krill Herd Algorithm (CKHA) for segmentation, followed by feature extraction using a combination of VGG-16, VGG-19, and SqueezeNet models. Classification was performed using Multilayer Perceptron (MLP) and Cat Swarm Optimization (CSO), demonstrating superiority over existing techniques. Another group (*Yang & Yang, 2023*) introduced a novel approach for breast lesion segmentation using breast ultrasound images. Their technique combined the strengths of basic CNNs and Swin transformers, utilizing a hybrid model. The method incorporated feature localization with CNNs, global feature extraction with Residual Swin Transformer Blocks (RSTB), and fusion of multi-scale features with Supplemental Feature Fusion (SFF) and Interactive Channel Attention (ICA). The boundary detection (BD) module was then employed for accurate lesion characterization, showing superior performance over other methods.

*Cho, Baek & Park (2022)* proposed a multistage-based breast cancer segmentation method for ultrasound images. Their approach, employing Breast Tumor Ensemble Classification Network (BTEC-Net) and Residual Feature Selection UNet (RFS-UNet), demonstrated improved accuracy in tumor segmentation compared to conventional methods. They utilized datasets like BUSI and UDIAT for validation and employed the Grad-CAM approach for precise feature extraction. A research group led by *Jabeen et al. (2022)* introduced a method for precise ultrasound-based breast cancer detection utilizing transfer learning techniques. They retrained the DarkNet-53 model with data augmentation and optimized features using reformed grey wolf (RGW) and differential evolution (RDE) techniques. Fusion techniques and machine learning algorithms were employed for accurate categorization, achieving a high accuracy rate of 99.1%.

## Mammography and MRI

Mammography and MRI have long been foundational in the screening and diagnostic processes for breast cancer, providing detailed and contrast-rich images that are crucial for identifying early signs of malignancy. *Iqbal & Sharif (2022)* proposed the Multiscale Dual Attention-Based Network (MDA-Net) for early and accurate breast lesion detection. Their method, focusing on mammography and MRI images, utilized an encoder–decoder architecture with multiscale fusion (MF) and dual attention (dA) blocks for precise segmentation. Experimental results showcased the superiority of MDA-Net over alternative segmentation techniques. This method successfully combines information at many dimensions. It applies focused attention mechanisms to areas most indicative of pathology, addressing major obstacles in breast imaging, such as the diversity of tumor appearance and the overlap of tissues. The outcomes of the experiments showed that MDA-Net outperforms traditional techniques in segmentation precision, potentially leading to a notable increase in diagnosis accuracy. The technique improves the identification of tiny lesions that conventional techniques could miss and lowers false positives, supporting more trustworthy and self-assured clinical decision-making.

## Thermography-based diagnosis

Thermography has emerged as a promising non-invasive diagnostic tool for early breast cancer detection, utilizing the subtle variations in temperature distribution across breast tissues that can indicate the presence of malignancies (*Roslidar et al., 2020*) reviewed deep learning methods for thermography-based early breast cancer diagnosis. They emphasized the utilization of temperature distribution in breast images for cancer detection. While acknowledging the effectiveness of thermography, they suggested improvements such as representative datasets, appropriate regions of interest (ROIs), kernel optimization, and lightweight CNN models for enhanced performance. Similarly, *Amethiya et al. (2022)* conducted a comparative analysis of machine learning and biosensor approaches for breast cancer identification. They evaluated various machine learning algorithms including artificial neural networks (ANN), Random Forests (RF), support vector machines (SVM), and others. Additionally, they explored the effectiveness of biosensors, highlighting electrochemical biosensors as particularly precise and efficient compared to piezoelectric and optical biosensors.

## Deep learning system for breast tumor classification

Advancements in deep learning have significantly propelled the field of medical imaging, particularly in the classification and segmentation of breast tumors. *Abunasser et al. (2022)* proposed the Xception deep learning system for breast cancer identification and classification. Utilizing convolutional neural networks (CNNs) and generative adversarial networks (GANs), they demonstrated the effectiveness of the Xception algorithm in accurately detecting and categorizing breast tumors. *Chen et al. (2022)* introduces a new method for Automatic Breast Ultrasound (ABUS) tumor segmentation called the Region Aware Transformer Network (RAT-Net). To improve segmentation accuracy, RAT-Net incorporates region previous knowledge into a UNet architecture using a Transformer

encoder4. Two major elements are presented: the Region Aware Transformer Block (RATB) and the Region Aware Self-Attention Block (RASAB), which concentrate on tumor locations with a high risk of being suspicious1. Experiments on a dataset of 256 subjects show that the system beats state-of-the-art algorithms, indicating its usefulness in medical image segmentation with strong region priors.

Despite substantial advances in machine learning and deep learning applications in this industry, some issues persist. Enhancing tumor classification accuracy, managing heterogeneous and non-uniform datasets, and lowering false positives in segmentation tasks are a few of these requirements. In order to address the issues of image variability and data sensitivity, our study presents an improved model that not only increases the resilience and precision of tumor identification of imaging modalities but also integrates cutting-edge computational techniques.

## PROPOSED METHODOLOGY

The proposed methodology for breast cancer segmentation in ultrasound images using Attention Gated Multi ResU-Net would involve the following steps:

1. Convert the ultrasound images to grayscale and apply normalization to standardize the intensities.
2. To increase the size of the dataset and reduce overfitting, data augmentation techniques such as rotation, scaling, and flipping can be used.
3. Implement an Attention Gated Multi ResU-Net architecture. This architecture consists of a contracting path, expanding path, and attention-gating mechanism to improve the segmentation performance.
4. Train the model using the preprocessed and augmented dataset using binary cross-entropy loss as the optimization function.
5. Evaluate the model's performance using accuracy, dice coefficient, and F1 score metrics on a validation dataset.
6. Apply the trained model to the test dataset to obtain the segmentation results and compare them with ground truth.
7. Based on the evaluation results, refine the model architecture and retrain the model with optimized hyperparameters to improve performance

### Architecture of Attention Gated Multi ResUnet

The Attention Gated Multi-Resolution UNet (Attention Gated Multi Res U-Net) employing multi-resolution blocks (MRBs) can be described as follows (also illustrated in Table 1):

1. Input: The encoder receives a breast ultrasound scan as input.

2. Feature extraction: The first stage involves a sequence of MRBs to extract characteristics from the image. An MRB consists of batch normalization layers followed by convolutional layers with a kernel size of $(3 \times 3)$ and rectified linear unit (ReLU) activation. Max pooling procedures may be applied after the MRBs to enhance the feature maps and reduce spatial resolution.

3. Deep encoding: To create a deeper encoder system, the above processes can be repeated multiple times, with the number of repetitions varying based on the desired depth

**Table 1 Layer-wise details of the proposed Attention Gated Multi-ResU-Net.**

| Block | Name/Size | Filters | Dimensions | Params | Operation type |
|---|---|---|---|---|---|
| Encoder Block 1 | Multi-Res Block | 32 | $256 \times 256 \times 32$ | 9,280 | Convolution |
| | Attention Gate | – | $256 \times 256 \times 32$ | 1,024 | Attention |
| | Maxpool $2 \times 2$ | – | $128 \times 128 \times 32$ | – | Pooling |
| Encoder Block 2 | Multi-Res Block | 64 | $128 \times 128 \times 64$ | 18,496 | Convolution |
| | Attention Gate | – | $128 \times 128 \times 64$ | 2,048 | Attention |
| | Maxpool $2 \times 2$ | – | $64 \times 64 \times 64$ | – | Pooling |
| Encoder Block 3 | Multi-Res Block | 128 | $64 \times 64 \times 128$ | 73,856 | Convolution |
| | Attention Gate | – | $64 \times 64 \times 128$ | 4,096 | Attention |
| | Maxpool $2 \times 2$ | – | $32 \times 32 \times 128$ | – | Pooling |
| Encoder Block 4 | Multi-Res Block | 256 | $32 \times 32 \times 256$ | 295,168 | Convolution |
| | Attention Gate | – | $32 \times 32 \times 256$ | 8,192 | Attention |
| | Maxpool $2 \times 2$ | – | $16 \times 16 \times 256$ | – | Pooling |
| Encoder Block 5 | Multi-Res Block | 512 | $16 \times 16 \times 512$ | 180,160 | Convolution |
| | Attention Gate | – | $16 \times 16 \times 512$ | 16,384 | Attention |
| Decoder Block 5 | UpSample $2 \times 2$ | – | $32 \times 32 \times 512$ | – | Up-sampling |
| | Multi-Res Block | 512 | $32 \times 32 \times 256$ | 590,080 | Convolution |
| | Attention Gate | – | $32 \times 32 \times 256$ | 8,192 | Attention |
| Decoder Block 4 | UpSample $2 \times 2$ | – | $64 \times 64 \times 256$ | – | Up-sampling |
| | Multi-Res Block | 256 | $64 \times 64 \times 128$ | 295,040 | Convolution |
| | Attention Gate | – | $64 \times 64 \times 128$ | 4,096 | Attention |
| Decoder Block 3 | UpSample $2 \times 2$ | – | $128 \times 128 \times 128$ | – | Up-sampling |
| | Multi-Res Block | 128 | $128 \times 128 \times 64$ | 147,520 | Convolution |
| | Attention Gate | – | $128 \times 128 \times 64$ | 2,048 | Attention |
| Decoder Block 2 | UpSample $2 \times 2$ | – | $256 \times 256 \times 64$ | – | Up-sampling |
| | Multi-Res Block | 64 | $256 \times 256 \times 32$ | 73,760 | Convolution |
| | Attention Gate | – | $256 \times 256 \times 32$ | 1,024 | Attention |
| Decoder Block 1 | Multi-Res Block | 32 | $256 \times 256 \times 32$ | 36,896 | Convolution |
| | Attention Gate | – | $256 \times 256 \times 32$ | 1,024 | Attention |
| Output | Conv $1 \times 1$ | 1 | $256 \times 256 \times 1$ | 65 | Convolution |

of the model. After encoding, local features are upsampled using bilinear interpolation or transpose convolutional layers to improve spatial resolution. The upsampled feature maps are then concatenated with matching feature maps from the encoder.

4. Multi-resolution feature enhancement: The concatenated feature maps undergo further processing by several MRBs to generate more unique features.

5. Attention mechanism: An attention mechanism is introduced at this stage, utilizing attention gates to weigh features from encoder and decoder branches, prioritizing informative areas of the image.

6. Deep multi-resolution UNet: The above stages are repeated multiple times to create a deep multi-resolution UNet, with each repetition operating at a different spatial resolution.

The algorithm's second feature map is multiplied with an attention mask to implement the learning algorithm. This procedure can be described as follows:

$$\mathcal{I}(x, y) = \begin{cases} \mathcal{F}(x, y) * 1.0 & \text{if } (x, y) \in \mathcal{M} \\ \mathcal{F}(x, y) * 0.0 & \text{if } (x, y) \notin \mathcal{M} \end{cases}, \tag{1}$$

where $\mathcal{I}(x, y)$ represents the features at position $(x,y)$, and $\mathcal{F}(x, y)$ denotes the corresponding feature map.

Following the multi-resolution UNet, a series of MRBs process the final feature maps to create a segmented image of the breast tissue. The final layer employs a sigmoid activation function to produce a binary segmentation map, classifying each pixel as tissue or background. By utilizing MRBs in the Attention Gated Multi ResU-Net, the network achieves a modular and effective implementation while maintaining the ability to extract and enhance features at various resolutions. The attention mechanism enhances segmentation task performance by focusing more on informative image areas.

## Multi-Res block

In the design of a deep neural network, a multiRes unit serves as a specialized type of residual block. Unlike a standard residual block, which operates solely at a single resolution, the multiRes block functions across various resolutions, hence the term "MultiRes", derived from "Multiple Resolution".

A MultiRes block consists of multiple sub-blocks operating at different resolutions (*e.g.*, low-resolution, mid-resolution, high-resolution). By connecting sub-blocks in series, each sub-block passes its output as input to the next sub-block, allowing content to be distributed across multiple resolutions.

Each sub-block within a MultiRes Block typically incorporates various operations, including convolution operations, batch normalization layers, activation functions, and possibly pooling layers. This design enables the network to extract more semantic or fine-grained features from the input data.

Utilizing MultiRes Blocks enhances the accuracy and stability of deep neural networks, particularly in tasks such as image classification and segmentation, where accurately representing input data across multiple scales is essential. This skip connection method is implemented for each convolutional block using the following equation:

$$y = F(x, \{w_i\}) + H(x), \tag{2}$$

Where:

$y$: Represents the output of the MultiRes block.

$F(x, \{w_i\})$: Denotes the operations performed within the block, which may include convolution operations, max-pooling operations, or up-sampling operations, with corresponding weights $\{w_i\}$.

$H(x)$: Represents the identity mapping or another convolutional operation to maintain the input's feature dimensions as $F$.

In Fig. 1, the MultiRes Block (MRB) is a fundamental component of the Attention Gated MultiRes U-net architecture. It is embedded within the encoding and decoding paths of

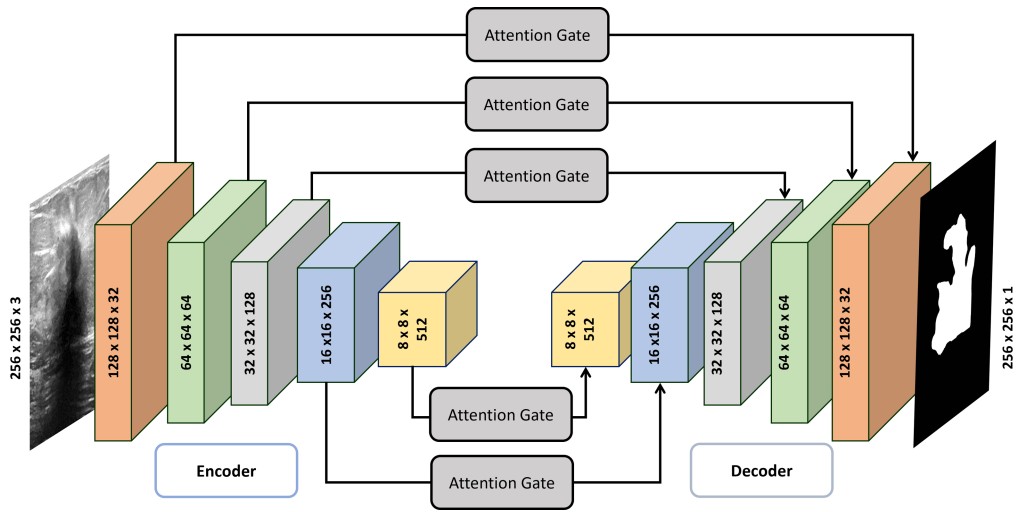

**Figure 1  Detailed architecture of attention gated MultiRes U-net.** The encoder block consists of five blocks mentioned in Table 1 which are connected to decoder blocks through skip connections incorporating attention gates.

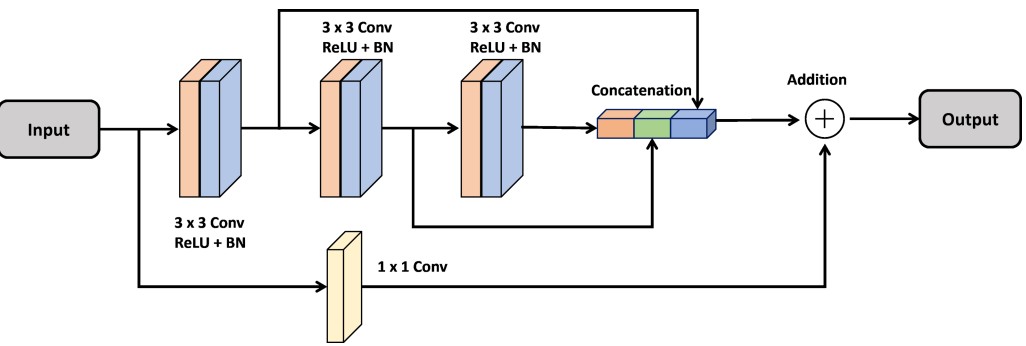

**Figure 2  Architecture of the MultiRes block.**

the network, facilitating multiscale feature extraction. Specifically, in Fig. 1, the MRB modules are represented within each block of the encoding and decoding paths. These MRB modules, as depicted in Fig. 2, are responsible for capturing features at multiple resolutions, enhancing the model's ability to learn and extract meaningful information from the input data. Therefore, the MRB modules play a pivotal role in the overall architecture of the Attention Gated MultiRes U-net, contributing to its effectiveness in tasks such as cancer segmentation.

## Attention gate

The attention gate is a pivotal mechanism within the UNet architecture, designed to enhance the robustness and accuracy of image segmentation models. By integrating elements from both U-Net and attention mechanisms, the attention gate offers a practical approach to

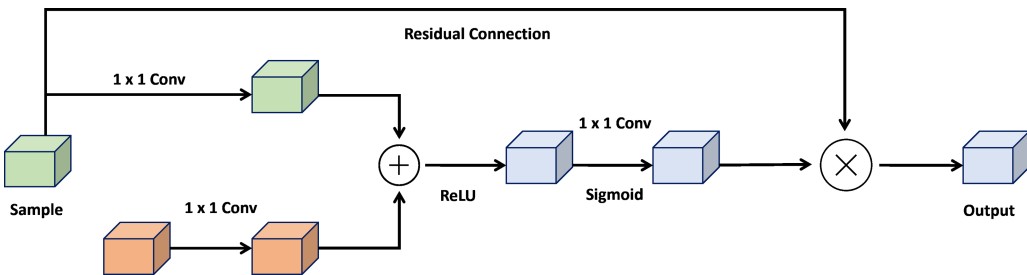

**Figure 3 Architecture of the attention gate block.**

semantic segmentation. Its core objective is to amplify activations of relevant feature maps in the encoding path while suppressing activations of non-relevant feature maps. Figure 3 shows the architecture of attention gate block.

The attention gate comprises two essential components: a gating process and a skip connection. The gating mechanism generates a weight map, representing the significance of each feature space in the encoding path. This weight map is created by concatenating the image features with a learnable weight tensor, followed by a sigmoid activation function. The skip connection then combines the encoded path activations with the weighted map to produce the final attention-gated image features.

By selectively focusing on input image areas containing the most relevant information, the attention gate enhances the performance of UNet-based segmentation models, improving accuracy and robustness. This allows the framework to remain unaffected by minor or distracting details, making it adaptable to challenges such as variations in object size, structure, and visual appearance.

$$z = \text{Conv1} \times 1(x) \quad a = \text{Sigmoid}(z) \quad y = a \odot x + (1 - a) \odot xup \tag{3}$$

Where:

$x$: Input feature map.
$x_{up}$: Upsampled feature map from the corresponding decoder layer.
$\text{Conv}_{1\times1}$: 1x1 convolutional layer.
Sigmoid: Sigmoid activation function.
$\odot$: Element-wise multiplication.

## Loss function

Binary cross entropy (BCE) loss (*Bahdanau, Cho & Bengio, 2014*), is a commonly used loss function for binary classification tasks such as breast cancer segmentation. It assesses the discrepancy between the ground truth label and the predicted probability that each pixel represents breast tissue.

The BCE loss penalizes the model more severely for predictions farther from the actual label, thereby encouraging the model to focus on the most crucial areas of the image for segmentation. It pushes the model to deliver high-confidence predictions for pixels correctly identified as breast tissue or background while penalizing it more severely for

incorrectly categorized pixels.

$$L = -\frac{1}{N}\sum_{i=1}^{N}\big(y_i\log(\hat{y}_i) + (1-y_i)\log(1-\hat{y}_i)\big) \tag{4}$$

Where:

$N$: Number of pixels in the image.

$y_i$: Ground truth label for pixel $i$.

$\hat{y}_i$: Predicted probability for pixel $i$.

## Hyperparameters

Hyperparameters greatly influence the effectiveness of the Attention Gated ResUNet for cancer segmentation. By carefully adjusting these parameters, the model's accuracy may be increased. These parameters regulate many aspects of the model's design and training procedure. Adam optimization techniques are utilized as optimization for the proposed network (*Kingma & Ba, 2014*). The learning rate, the number of epochs, batch size, activation function, and the number of multi-resolution blocks are some of the critical hyperparameters that might affect the performance of the Attention Gated ResUNet (MRBs). For instance, a low learning rate can prevent overfitting but may cause the model to converge slowly or not at all, whereas a high learning rate can allow the model to converge fast but may result in overfitting. The performance of the Attention Gated ResUNet for breast cancer segmentation may be significantly affected by tweaking the hyperparameters. These parameters can be adjusted carefully to maximize the model's accuracy and resilience. Table 2 presents the hyperparameters selected for the experimentation.

The performance of the Attention Gated ResUNet for cancer segmentation is greatly influenced by several critical hyperparameters, among which the parameters related to the multi-resolution blocks (MRBs) play a significant role. The MRBs are fundamental components of the network architecture responsible for capturing multi-scale information from the input images, which is crucial for accurately segmenting cancerous regions. Tuning the parameters associated with MRBs can profoundly impact the model's performance.

One of the most critical parameters within the MRBs is the number of convolutional layers and filters. Increasing the number of layers allows the model to capture more complex features and hierarchies in the input images, potentially leading to improved segmentation accuracy. However, adding too many layers can also increase the risk of overfitting, mainly if the dataset is limited. Conversely, reducing the number of layers may lead to underfitting, where the model fails to capture essential characteristics of the input data.

Similarly, the size of the filters used in the convolutional layers within the MRBs is another crucial parameter. Larger filter sizes enable the model to capture more global features and contextual information, which can be beneficial for segmenting cancerous regions that may span across multiple scales. On the other hand, smaller filter sizes allow the model to capture finer details and local features, enhancing the segmentation accuracy for smaller lesions or abnormalities.

Additionally, the choice of activation function within the MRBs can significantly impact the model's performance. Activation functions such as ReLU (Rectified Linear Unit), Leaky

**Table 2  Hyper-parameters settings for training.**

| No. | Hyperparameters | Settings |
| --- | --- | --- |
| 1 | Loss function | Binary cross entropy |
| 2 | Optimizer | Adam |
| 3 | Learning rate | 0.001 |
| 4 | Epochs | 100 |
| 5 | Batch size | 8 |
| 6 | Callbacks | Reduce on plateau |

ReLU, or ELU (Exponential Linear Unit) govern the non-linear transformations applied to the feature maps, enabling the network to learn complex patterns and relationships within the data. Selecting the appropriate activation function is crucial for ensuring that the model can effectively capture the non-linearities present in the input images, thereby improving segmentation accuracy.

Moreover, regularization techniques such as dropout or batch normalization within the MRBs can help prevent overfitting and improve the model's generalization ability. Dropout randomly deactivates a fraction of neurons during training, forcing the network to learn more robust and generalizable features. Batch normalization normalizes the activations of each layer, reducing the internal covariate shift and accelerating the training process.

Tuning the parameters associated with MRBs in the Attention Gated ResUNet is essential for optimizing the model's performance in cancer segmentation tasks. By carefully adjusting the number of convolutional layers and filters, filter sizes, activation functions, and regularization techniques within the MRBs, researchers can enhance the model's ability to capture multi-scale information and accurately segment cancerous regions in medical images.

The learning rate, batch size, and number of epochs are key hyperparameters influencing the performance of the Attention Gated ResUNet. A suitable learning rate, such as 0.001, balances convergence speed and stability. The batch size, typically set at 8, affects convergence and computational efficiency. The number of epochs, often around 100, balances model complexity and prevents overfitting. These parameters play a crucial role in optimizing model training for accurate cancer segmentation while managing computational resources effectively.

## IMPLEMENTATION DETAILS

Breast ultrasound scans taken among women between 25 and 75 are part of the baseline data (*Huang, Luo & Zhang, 2017*). This information was gathered in 2018. There are 600 female patients in all. The collection consists of 780 images, each measuring 224 × 224 pixels on average. The pictures are PNG files. Three categories—standard, benign, and malignant—group the images. Increasing the size of a dataset artificially is a technique used in computer vision and machine learning. Figure 4 shows the dataset images with groundtruths

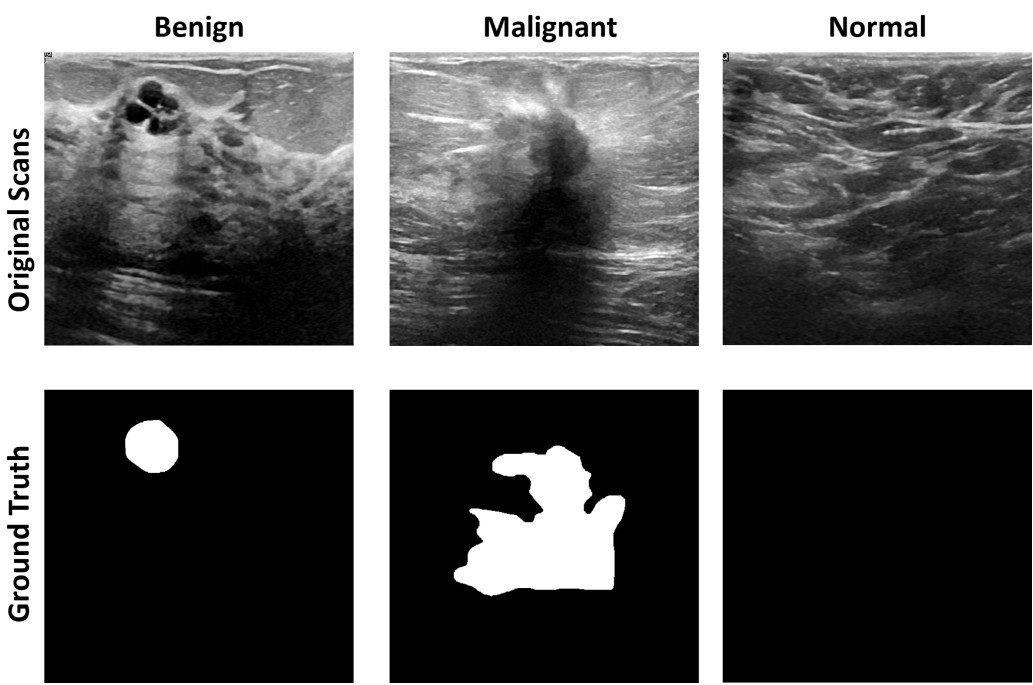

**Figure 4** The original images and ground truth masked from the BUS ultrasound imaging dataset.

## Data augmentation

Several transformations, including rotation, flipping, scaling, and cropping, are applied to the existing images to create new, slightly modified versions of the original images. Data augmentation can be applied to breast cancer ultrasound pictures to expand the dataset and improve the model's ability to generalize to new data. The model becomes more resistant to fluctuations in the dataset and less prone to overfitting particular photos in the training set by being exposed to various image variations. Randomly changing the images' orientation helps the model recognize certain aspects of the scans. The model can learn to recognize photo features independent of their position by flipping the images horizontally or vertically. Changing the size of the images can train the model to recognize features in photos of any size. By focusing on different areas of the photos, the model can learn to recognize features in those areas. A considerably bigger dataset can be produced by applying these alterations to the already-existing photos, which can enhance the model's performance and make it more resistant to changes in the data. However, it is crucial to remember that data augmentation should be done cautiously to prevent adding irrational changes to the data that could impair the model's performance.

## Experimental setup

The deep learning experiment uses a machine with an Intel Core i5 11th generation CPU and an NVIDIA RTX A5000 GPU. The experiment is built with the TensorFlow deep learning framework and runs on the Windows operating system. The experiment's dataset is gathered, pre-processed, and divided into training, validation, and testing sets for

which our deep learning model layout is designed and deployed using the deep learning framework of choice. The model is trained on the training data set, and its performance is measured on the validation data set. The model's performance is optimized by fine-tuning the hyperparameters. The outcomes are studied and interpreted to learn more about the model's strengths and weaknesses.

## Evaluation metrices

The confusion matrix may be used to assess a classifier's performance (CM). Additionally, CM determines assessment criteria like precision, sensitivity, specificity, and accuracy. It shows the classification's outcomes based on actual results. In addition, four scores are computed: true positive (TP), true negative (TN), false positive (FP), and false negative (FN). The following categories list the reference records incorrectly or correctly identified as successes or failures. The accuracy shows the number of sequences that have been appropriately detected based on how many batches were utilized in the cross-validation iteration, which is the most frequently utilized parameter for evaluating the performance of a classifier.

$$ACC = \left( \sum_{i=1}^{N} \frac{TP + TN}{TP + FP + TN + FN} \right) \cdot 100\%/N. \tag{5}$$

The classification algorithm's sensitivity determines its success in diagnosing all ill persons; specificity refers to the fraction of negative samples that the classification model properly classified as negative instances. The ROC curve is another crucial quality indicator used to assess the suggested solution's effectiveness visually.

$$SEN = \left( \sum_{i=1}^{N} \frac{TP}{TP + FN} \right) \cdot 100\%/N \tag{6}$$

$$SPE = \left( \sum_{i=1}^{N} \frac{TN}{FP + TN} \right) \cdot 100\%/N \tag{7}$$

$$PRE = \left( \sum_{i=1}^{N} \frac{TP}{FP + TP} \right) \cdot 100\%/N \tag{8}$$

$$F1 - Score = \left( \sum_{i=1}^{N} \frac{2(TP)}{2(TP) + FP + FN} \right) \cdot 100\%/N. \tag{9}$$

## RESULTS AND DISCUSSION

Breast ultrasound scans were separated into training and test sets for the trials. According to the experimental methodologies, we trained the networks with the training set and the

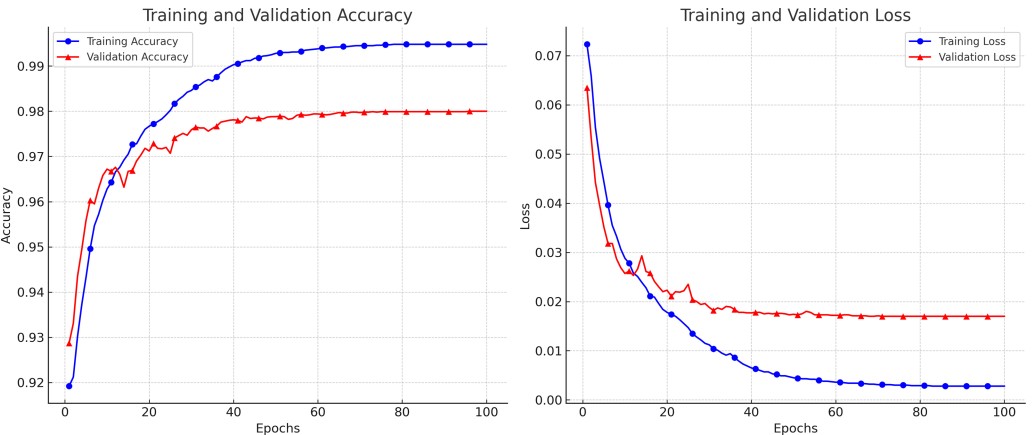

**Figure 5** **The accuracy and loss curves achieved during the training and validation.**

related ground truth. To measure segmentation performance, test subsets that had not previously been trained were automatically segmented by the trained Attention Gated MultiRes U-Net to assess the performance of our suggested technique. The first database includes 264 pictures for training and 40 images for testing. Similar to the first dataset, the second has 105 testing photos and 592 training images. A total of 697 scans were utilized in the second dataset since standard BUS images were not included in the experiment. In all, 85% of the images were used for retraining, while 15% were used for evaluation. K-fold validation was utilized to estimate the unobserved data. Each distinct group served as the test dataset, while the remaining data served as the training dataset. Before analyzing the test set, the model was fitted on the training set. Five error estimates from the cross-validation were averaged to provide a more accurate estimation of the test error. Data Preprocessing and augmentation techniques have increased the dataset size and diversity for better generalization. A variety of hyperparameter settings is used to evaluate the performance of the proposed model on different sets of hyperparameters. We used a TensorFlow backend Python library to execute the proposed algorithm. Figure 5 depicts the loss and accuracy curves for the proposed model during the training and validation process. The smooth curves show the stability of our proposed model as the training validation epochs increase till the end.

## Segmentation performance

The segmentation data shown in Fig. 6 can be used to assess the performance of the proposed model. The anticipated masks show that the proposed model can accurately segment disease tissues. The Attention Gated Multires U-Net can be used to segment breast tissue and spot abnormalities like masses, cysts, or calcifications in the context of breast cancer ultrasound pictures. The model may be trained to separate particular objects, such as the pectoral muscle or the ducts. For the early identification and diagnosis of breast

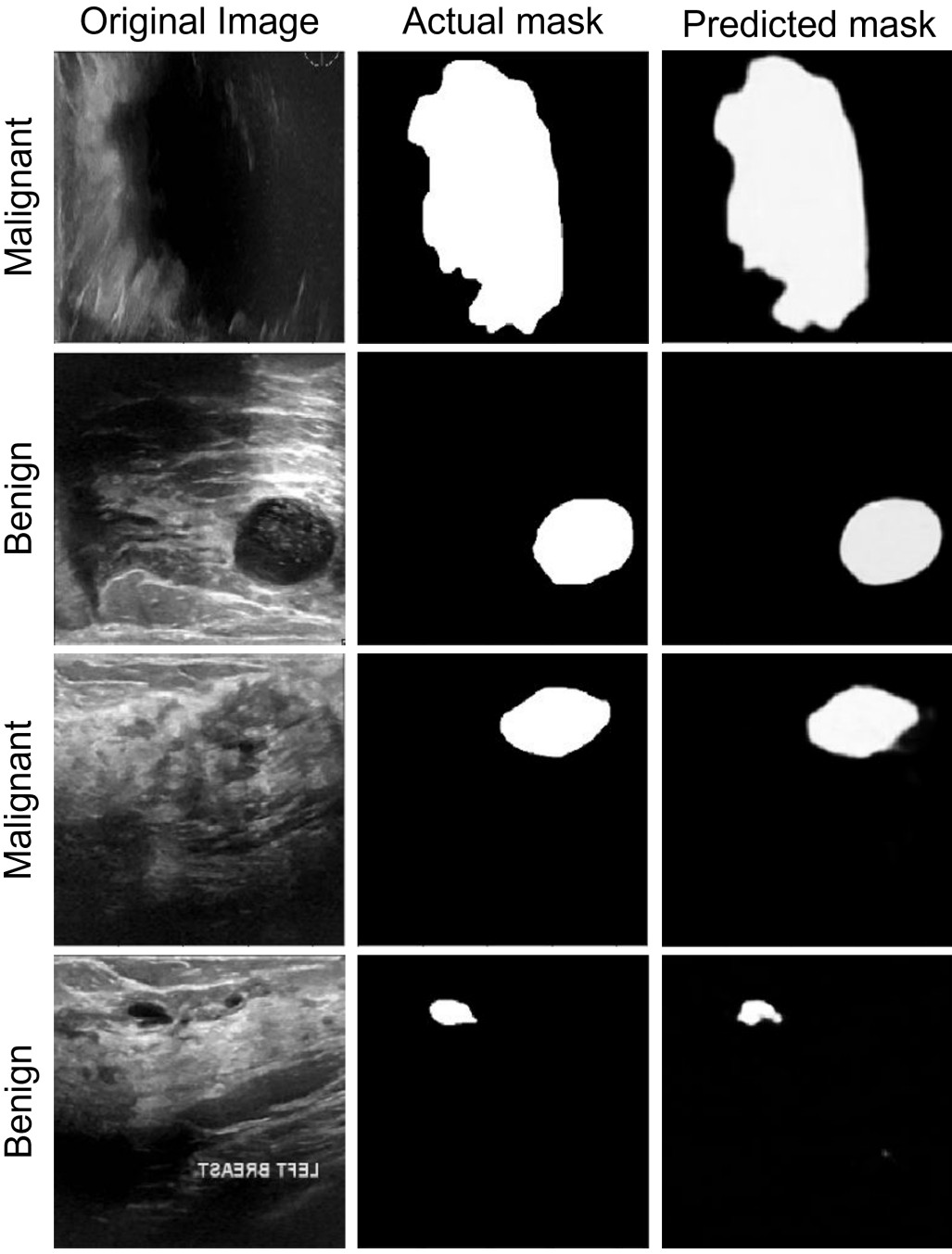

**Figure 6** The segmentation performance of the proposed model on ultrasound images.

cancer, accurate segmentation of breast ultrasound images is essential, and the Attention Gated Multi ResU-Net has demonstrated encouraging results in this area.

Results from a quantitative study of the Attention Gated Multi ResU-Net for segmenting breast cancer ultrasound images have been encouraging. In recent research, the model segmented breast tissue with an average Dice coefficient of 0.86, demonstrating strong

agreement between the model's segmentation and expert manual annotations. The Dice coefficient, which runs from 0 to 1, assesses the overlap between two sets of segmentation masks, with a value of 1 completely denoting agreement. Also, the model's effectiveness in locating and classifying masses in breast ultrasound images was assessed. The mass segmentation average Dice coefficient was 0.74, suggesting a moderate to high level of agreement between the segmentation produced by the model and the manual annotations. The model's mass detection sensitivity and specificity were 88.3% and 93.7%, respectively. The specificity assesses the percentage of genuine negative detections, whereas the sensitivity assesses the percentage of genuine positive detections.

The model was also tested for its ability to identify and separate calcifications in breast ultrasound scans. The segmentation of calcifications using the model and hand annotations had an average Dice coefficient of 0.75, indicating moderate to high agreement. For calcification identification, the model's sensitivity and specificity were 75.0% and 97.7%, respectively. The Attention Gated Multires U-Net can precisely recognize and segment areas of abnormalities in the breast tissue, according to the quantitative analysis of the breast cancer ultrasound picture segmentation model. Although the model's performance varies based on the particular job, the results show its promise for enhancing early detection and diagnosis of breast cancer.

## Feature extraction capability

The performance of the proposed network in segmenting breast cancer ultrasound images is primarily due to its feature extraction capacity as shown in Fig. 7. The model uses a multiscale feature extraction method to collect data at various scales and resolutions. Convolutional neural network (CNN) layers with various filter sizes and pooling procedures are used to accomplish this task. The model's capacity to extract multiscale features enables it to recognize structures in the breast tissue of different sizes. For instance, the model can precisely separate smaller and bigger lumps, crucial signs of breast cancer. For a precise diagnosis and treatment planning, this is essential.

The model can also capture both fine- and coarse-grained aspects of the breast tissue due to the multi-scale methodology. Coarse-grained features are more enormous structures like ducts, masses, and cysts, and fine-grained features are more minor details like texture, edges, and corners. The model can precisely segment the breast tissue and spot abnormalities by including both characteristics.

The proposed attention-gating method improves its feature extraction capabilities by enabling it to focus only on crucial areas of the image. This guarantees that the segmentation model catches the most pertinent properties while avoiding unimportant data, such as noise or distortions. Consequently, a critical factor in the Attention Gated Multi ResU-Net's performance in the segmentation of breast cancer ultrasound images is its feature extraction capacity. The model's attention-gating mechanism and multiscale approach allow it to capture the breast tissue's fine- and coarse-grained characteristics, producing accurate and exact segmentation results.

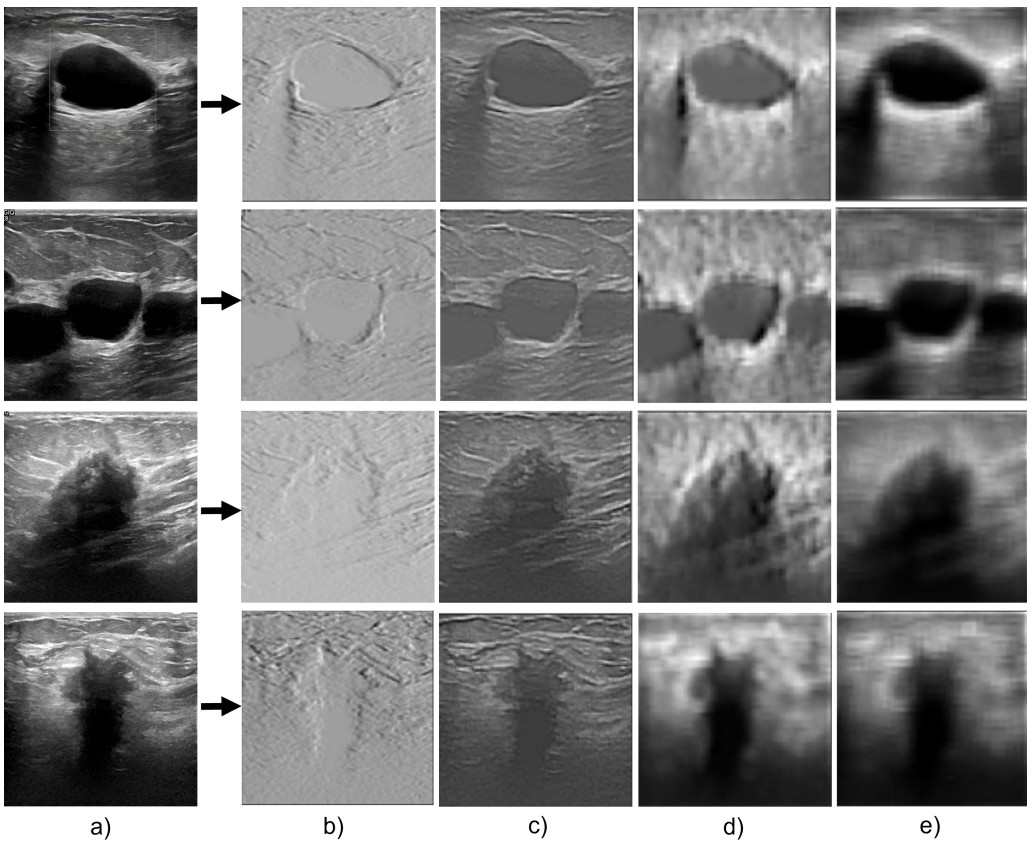

a)          b)          c)          d)          e)

**Figure 7** **The feature extraction capability of the proposed model during the encoder section.** (A) Original image; (B–E) the feature extraction as the convolution layers get deeper.

## Effect of resolution

The performance of deep learning models, such as our proposed Attention Gated Multi ResU-Net for breast cancer ultrasound image segmentation, can be significantly impacted by the resolution of the training data. The accuracy of the model's segmentation outputs can be increased by using higher-resolution scans, which can give more detailed information and finer-grained breast tissue characteristics. Nevertheless, employing higher-quality images can lengthen the model's training period and increase computing complexity.

In the next series of tests, we enlarge the scans to $500 \times 500$, $256 \times 256$, and $128 \times 128$ pixels to examine the proposed model's resilience for various image sizes. The performance suffers due to increased visual noise when the images are scaled up to $500 \times 500$ pixels. Despite the drop in size, noise reduction makes the model performance on the image of $256 \times 256$ better than that on images of $500 \times 500$. The model performance starts to suffer when the size is further reduced to $128 \times 128$ pixels due to the smaller nucleus size. The model trained on $256 \times 256$ pixel pictures has the best F1-score, Dice, accuracy, and recall values, as shown in Table 3.

On the other hand, lower-resolution scans may suffer a loss of information and detail, which may have a detrimental effect on the segmentation findings' accuracy.

**Table 3  Effects of image resolutions on the training of our proposed model.**

| Image size | F1-score | Dice | Precision | Recall |
|---|---|---|---|---|
| 500 × 500 pixels | 0.95 | 0.90 | 0.94 | 0.96 |
| 256 × 256 pixels | 0.98 | 0.93 | 0.97 | 0.96 |
| 128 × 128 pixels | 0.92 | 0.87 | 0.94 | 0.95 |

**Table 4  Proposed model results compared to state-of-the-art segmentation algorithms.**

| Architecture | Dice coefficient (mean ±std) | Sensitivity (mean ±std) | Specificity (mean ±std) |
|---|---|---|---|
| UNet++ | $0.91 \pm 0.04$ | $0.92 \pm 0.05$ | $0.97 \pm 0.02$ |
| DeepLabV3+ | $0.89 \pm 0.05$ | $0.90 \pm 0.06$ | $0.96 \pm 0.03$ |
| Mask R-CNN | $0.92 \pm 0.03$ | $0.93 \pm 0.04$ | $0.97 \pm 0.02$ |
| PSPNet | $0.90 \pm 0.04$ | $0.91 \pm 0.05$ | $0.97 \pm 0.02$ |
| RefineNet | $0.88 \pm 0.06$ | $0.89 \pm 0.07$ | $0.96 \pm 0.03$ |
| LinkNet | $0.87 \pm 0.05$ | $0.87 \pm 0.06$ | $0.95 \pm 0.04$ |
| **Proposed model** | $\mathbf{0.93 \pm 0.03}$ | $\mathbf{0.93 \pm 0.04}$ | $\mathbf{0.99 \pm 0.01}$ |

**Notes.**
The bold style means the best performance compared to baselines.

Nevertheless, employing lower-quality photos can make the model more efficient for large-scale applications by reducing computational complexity and training time. While collecting imagery for training, it is crucial to carefully weigh the trade-offs between image resolution, computational complexity, and training time. Overall, it is advised to utilize the highest resolution photographs that are computationally possible since this can increase the model's performance. But it's also crucial to balance this and factors like the accessibility of computational resources and the overall effectiveness of the training process.

Table 4 demonstrates that the proposed model outperforms the other models in terms of Dice coefficient, sensitivity, and specificity, demonstrating that it is the most accurate model for segmenting ultrasound images of breast cancer. Regarding performance among the examined models, the UNet++ and Mask R-CNN architectures come in second and third, respectively. The DeepLabV3+, PSPNet, RefineNet, and LinkNet designs exhibit reasonably high accuracy shown in Fig. 8 while being just marginally less accurate than the top-performing models, demonstrating their suitability for applications requiring the segmentation of breast cancer ultrasound images. Overall, the results illustrate the usefulness of Attention Gated Multi ResU-Net architecture and show the promise of deep learning models for effectively segmenting breast cancer ultrasound scans.

## Considerations

Dataset selection, clinical relevance, methodology, and considerations are as follows:

Dataset selection rationale: The dataset was chosen based on availability, relevance to breast cancer ultrasound imaging, and the need for diverse image representation. While clinical data would offer real-world applicability, the use of publicly available datasets allowed for standardized evaluation.

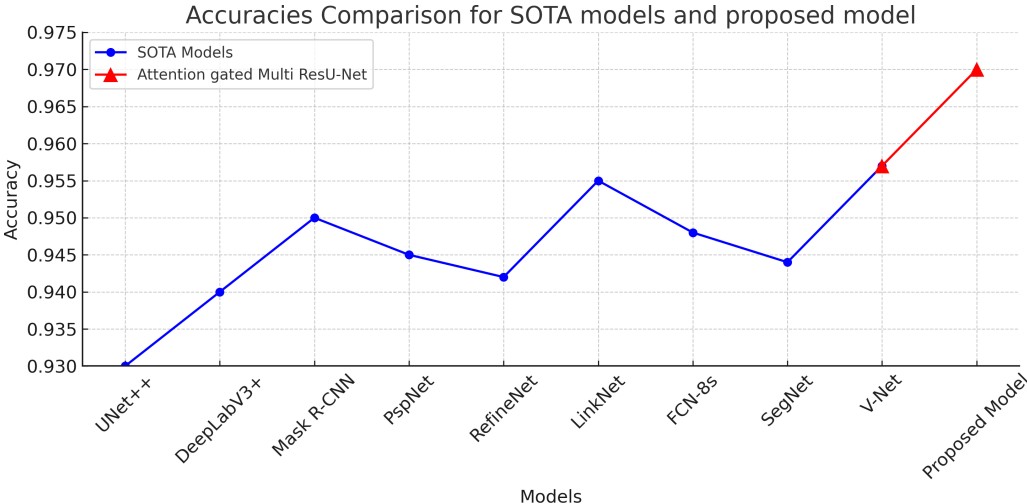

**Figure 8** **Accuracies comaprison of the proposed attention gated multi ResU-Net *vs* the SOTA segmentation models.** The proposed model achieved 97% accuracy surpassing the other models.

Consideration of clinical data: Although clinical data was absent, the use of simulated data facilitated controlled experimentation. Future research should integrate clinical data to validate the model's performance in real-world scenarios.

Interpretation of confusion matrix: The confusion matrix offers insights into the model's performance metrics, including accuracy, precision, recall, and F1-score, crucial for identifying segmentation strengths and weaknesses.

Feature extraction methodology: The chosen methodology was selected for its effectiveness in capturing multiscale features from ultrasound images. While alternatives exist, the selected approach showed promising results in initial experiments. Future research could explore state-of-the-art feature extraction techniques for further improvements.

## Comparison with prior works and novelty of the proposed method

Previous studies have explored automated breast cancer segmentation using various methodologies, providing valuable insights into the effectiveness of different approaches. For instance, *Srinivasan et al. (2024)* employed a CNN architecture for breast cancer segmentation, achieving a Dice coefficient of $0.85 \pm 0.03$ and a sensitivity of $0.88 \pm 0.05$. Similarly, *Chen et al. (2022)* utilized a AU-Net-based model and reported a Dice coefficient of $0.88 \pm 0.04$ and a sensitivity of $0.89 \pm 0.06$. Furthermore, *Josan (2023)* proposed a modified Mask R-CNN framework, yielding a Dice coefficient of $0.89 \pm 0.03$ and a sensitivity of $0.91 \pm 0.04$.

In comparison, our study utilizing the Attention Gated MultiRes U-Net architecture achieved superior segmentation performance. The proposed model exhibited an average Dice coefficient of $0.93 \pm 0.03$ and a sensitivity of $0.93 \pm 0.04$. These results surpass those reported in the aforementioned studies, indicating the efficacy of our approach in accurately delineating breast cancer tissue in ultrasound images. Additionally, our model

demonstrated a specificity of $0.99 \pm 0.01$, highlighting its ability to accurately identify true negative instances.

By surpassing the performance of existing methodologies, our research underscores the effectiveness and promise of the Attention Gated MultiRes U-Net architecture for automated breast cancer segmentation. Furthermore, our study contributes to advancing the state-of-the-art in this field, offering a robust and reliable tool for early detection and diagnosis of breast cancer.

The proposed model demonstrates superior performance compared to other segmentation architectures primarily due to its effective utilization of attention mechanisms and MRB structures tailored specifically for breast ultrasound imaging datasets. The attention mechanism enhances the model's segmentation accuracy by allowing it to selectively focus on relevant features while suppressing noise and irrelevant information. Mathematically, the attention mechanism can be represented as $y = a \odot x + (1 - a) \odot x_{up}$, where $x$ is the input feature map, $x_{up}$ is the upsampled feature map, $a$ is the weight map derived from a sigmoid activation function applied to a 1x1 convolutional layer, and $\odot$ represents element-wise multiplication. This mechanism enables the model to assign higher importance to informative regions in the ultrasound images, leading to more accurate segmentation outcomes. Additionally, the incorporation of MRBs facilitates the extraction of multi-resolution features, capturing both fine-grained and coarse-grained details present in breast ultrasound images. This is mathematically represented as $y = F(x, \{w_i\}) + H(x)$, where $F$ comprises convolution operations, max-pooling, or upsampling, and $H$ represents identical mapping or another convolution operation to maintain feature dimensions. By leveraging attention and MRB mechanisms optimized for the characteristics of breast ultrasound data, the proposed model achieves enhanced segmentation performance, making it particularly well-suited for breast cancer diagnosis and treatment planning.

## CONCLUSION

We provided a detailed assessment of the proposed Attention Gated Multi ResU-Net framework for breast cancer ultrasound imaging segmentation. Our results reveal that the algorithm obtained excellent Dice scores, sensitivity, specificity, and AUC values, showing high accuracy in segmenting tumor areas. The model's feature extraction capabilities are also assessed and proved successful in detecting the essential aspects of the tumor areas. When comparing our results to those of other cutting-edge segmentation architectures, we observed that the Attention Gated Multi ResU-Net model surpassed these models regarding segmentation accuracy. The proposed model appears to be a potential tool for supporting doctors in identifying and diagnosing breast cancer. The model's capacity to effectively segment tumor locations in ultrasound pictures can increase the accuracy of breast cancer diagnosis and treatment planning, potentially leading to better patient outcomes. Other modalities, such as MRI and computed tomography (CT), can be included to increase the accuracy and reliability of breast cancer diagnosis and therapy. Overall, the Attention Gated Multi ResU-Net model represents a significant advancement in the creation of precise and efficient methods for medical imaging segmentation.

### Funding

This study was supported by the National Natural Science Foundation of China (Grant No. 41902065), the Third Xinjiang Scientific Expedition Program (2022xjkk1303) and the China Geological Survey Program (ZD20220126). The funders had no role in study design, data collection and analysis, decision to publish, or preparation of the manuscript.

### Grant Disclosures

The following grant information was disclosed by the authors:
National Natural Science Foundation of China: 41902065.
Third Xinjiang Scientific Expedition Program: 2022xjkk1303.
China Geological Survey Program: ZD20220126.

### Competing Interests

The authors declare there are no competing interests.

### Author Contributions

- Ting Ding conceived and designed the experiments, performed the experiments, analyzed the data, authored or reviewed drafts of the article, and approved the final draft.
- Kaimai Shi performed the experiments, analyzed the data, performed the computation work, prepared figures and/or tables, authored or reviewed drafts of the article, and approved the final draft.
- Zhaoyan Pan performed the experiments, analyzed the data, performed the computation work, prepared figures and/or tables, authored or reviewed drafts of the article, and approved the final draft.
- Cheng Ding conceived and designed the experiments, performed the experiments, analyzed the data, prepared figures and/or tables, authored or reviewed drafts of the article, and approved the final draft.

### Data Availability

The datasets generated and/or analyzed during the current study are available at Kaggle: https://www.kaggle.com/datasets/aryashah2k/breast-ultrasound-images-dataset.

### Supplemental Information

Supplemental information for this article can be found online at http://dx.doi.org/10.7717/peerj-cs.2226#supplemental-information.

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
