# Peer review of "AI-based automated breast cancer segmentation in ultrasound imaging based on Attention Gated Multi ResU-Net"

_PeerJ Computer Science, doi:10.7717/peerj-cs.2226_

## Round 0.1 · original submission · Major Revisions

Reviewers have suggested a major revision. You are required to revise your manuscript by considering all the comments and suggestions of the reviewers. The revised manuscript will be subjected to a 2nd round review.

**Language Note:** The review process has identified that the English language must be improved. PeerJ can provide language editing services - please contact us at [email protected] for pricing (be sure to provide your manuscript number and title). Alternatively, you should make your own arrangements to improve the language quality and provide details in your response letter. – PeerJ Staff

Reviewer 1 ·

Basic reporting

The manuscript lacks proper English and it is highly advised to complete the proof read with professional services / fluent English support speakers.

The Literature Review is not well organised and State of Art Section should be upgraded to Literature Review in an organised way.

The figures submitted by the authors are not clear and as per the Journal Guidelines. The figures should be modified for clarity of vision and understanding into vector graphics.

Several terms in the Formal results do not include the definitions of the terms used in the equations and theorems. It is highly recommended to update these formulae with proper terms and meaning.

Experimental design

The research focus on an AI-based automated breast cancer segmentation in Ultrasound Imaging based on Attention Gated Multi ResU-Net. However it is found that the ResU-Net is not clearly explained in terms of the conceptualisation and definition for the process involved. The proposed study seems to use the image processing, however, it lacks the novelty in research - i.e. a comparison study based on a dataset is done before as well by:

https://www.sciencedirect.com/science/article/abs/pii/S0301562920302878
https://www.sciencedirect.com/science/article/abs/pii/S0957417420306771
https://www.ncbi.nlm.nih.gov/pmc/articles/PMC7762151/
https://arxiv.org/ftp/arxiv/papers/2204/2204.12077.pdf
https://proceedings.mlr.press/v172/zhu22b/zhu22b.pdf
https://www.frontiersin.org/journals/oncology/articles/10.3389/fonc.2021.680807/full

The focus of the author should be more on comparison and its salient features identification instead of the model only.
Investigations are done using image processing, however it is recommended to use one more methods for the image analysis to add on.

Several questions remained unanswered such as : Why the particular dataset is chosen? Why no actual clinical data is taken into consideration? What is the nature of the confusion matrix in the study? Why feature extraction is accomplished in the way it is represented in the research - why not the state-of-the-art methodology is adapted?

Validity of the findings

The dataset used are clear but obsolete and legacy data. It is highly advised to the authors so that they can train the proposed study using some latest research data or some clinical dataset to perform the operation. Classification of the dataset can be done with tools like WEKA to make several classifiers work across the same. The results obtained then will be more beneficial.

Pixel resolution quality of the imaging dataset should be enhanced to a higher level. The legacy dataset fails to provide high resolution images for more clear understanding of the malignancies and accuracy in the parameters like F1 Score, Dice, Precision etc.

The results for the comparitive study with the state-of-the-art methods should be presented with graphical representation for better understanding of the comparison.

Segmented tumor area boundaries are indeed much more important during diagnosis. There is no substantial evidences in the study to prove that the segmentation is done in a proper manner covering all the possible areas. It is highly recommended to prove that the results for the identification of boundaries are well analysed by the model proposed.

A clear definition for attention gate should be presented in the study with proper reasoning for the choice of selection in this case.

Authors are advised to use the datasets given in :
https://pubmed.ncbi.nlm.nih.gov/31867417/
https://scholar.cu.edu.eg/?q=afahmy/pages/dataset

To validate their model and its results. They must include the validity after the application of several other datasets as well in the proposed model.

Additional comments

Several new citations and reference should be included in the manuscript.
Kindly make sure that the Table Citations and Figure Referencing appears in an ascending sequence in the manuscript.

All the figures under results section must explain the findings and then comparison with the previous models used for similar kind of study as per the literature review.

It is highly recommeded to upgrade the quality of images used in Fig. 4,5,6 etc. to vector graphics.

The paper contains potential but with the aforesaid changes it can be more beneficial and acceptable.
Authors are advised to complete the reviews and modify the findings as per the above sections.

Reviewer 2 ·

Basic reporting

The authors proposed an Attention-Gated Multires Unet model for medical image segmentation using a breast cancer dataset. This is one of the hot topics in medical imaging, utilizing deep learning, specifically employing Attention mechanisms. The originality of the proposed work lies in the features extraction and attention-gating mechanisms to improve performance in image segmentation. Overall, the manuscript is clear and easy to follow, but its current form is not ready to be published in this journal. Some improvements are needed as follows:

- Please check and use PeerJ formatting.
- Get the script proofread for English clarity, formatting, and grammar.
- Address the inconsistency in the names of the proposed method. For example, the title is "Attention Gated Multi ResU-Net," while line 22 reads "Attention Gated Multiures Unet," line 197 says "Attention Gated Multi ResUNet," and line 208 uses "AG-MR UNet." Authors should use one name consistently.
- In line 355, correct "5scan" to the appropriate term.
- Use an appropriate equation font, e.g., in line 278 for "yi."
- Recheck the overall structure and contents.
- Include the trend and direction in the last paragraph of the Related Works section.

Experimental design

- The methodology follows a standard process of the Attention mechanism. The author exclusively employs the Attention Gated MultiResU-Net in the architecture, while the remaining steps are part of the standard procedure.
- The author should provide related works on Multi-Resolution Blocks (MRBs) since this is a unique aspect of this attention scheme.
- Some confusion arises in Figure 1 until Figure 2. Clarify the location of the MRB (Figure 2) in Figure 1.
- Discuss more on the most critical hyperparameters that can significantly affect performance. For instance, elaborate on the impact of performance when tuning MRB parameters.
- In Table 1, the author should justify the usage of these parameters. Highlight only the most important hyperparameters that can affect performance.

Validity of the findings

- Line 343-344, provide the source of datasets
- Table 3 shows the result comparison. Author should justify why the proposed model outperforms others in terms of attention and MRB mechanism based on used breast ultrasound dataset.
- The finding is acceptable and shows a good knolwege extension, specifically using the Attention Gated Multires Unet model.

---

## Round 0.2 · Minor Revisions

Dear Authors,

Some minor changes are further required in order to proceed further:

1) In the abstract, the statement "The model's multi-scale feature extraction and attentiongating mechanism enable it to accurately identify and segment areas of abnormality in the breast tissue, such as masses, cysts, and calcifications." is sufficient. You need to mention some statistical results related to obtained level of accuracy.
2) Line no. 482, "Consideration" is unclear. Authors need to elaborate this section.
3) The limitations and de-limitations of the proposed method and future work to rule out those limitations and de-limitations need to be mentioned in the "Conclusion" section.

Reviewer 1 ·

Basic reporting

The authors have resolved all the observations and the manuscript is well revised.

Experimental design

The authors have resolved all the updates in the experimental design and the manuscript is well revised.

Validity of the findings

The validity of findings in the manuscript is now suitable as per the ethical considerations.

Additional comments

The authors have resolved all the observations and the manuscript is well revised.
The manuscript is suitable to be published with the recent revision.

---

## Round 0.3 · accepted · Accept

I am happy to accept your manuscript for the publication in the current form. Your manuscript under went 2 Rounds of critical review before coming to the conclusion.